# A Cell-Penetrating Peptide Modified Cu_2−x_Se/Au Nanohybrid with Enhanced Efficacy for Combined Radio-Photothermal Therapy

**DOI:** 10.3390/molecules28010423

**Published:** 2023-01-03

**Authors:** Ruixue Ran, Sinan Guo, Xiaoyu Jiang, Zhanyin Qian, Zhaoyang Guo, Yinsong Wang, Mingxin Cao, Xiaoying Yang

**Affiliations:** 1Tianjin Key Laboratory of Technologies Enabling Development of Clinical Therapeutics and Diagnostics (Theranostics), School of Pharmacy, Tianjin Medical University, Tianjin 300070, China; 2School and Hospital of Stomatology, Tianjin Medical University, Tianjin 300070, China

**Keywords:** radiosensitizer, TAT peptide, radiotherapy, photothermal therapy, combined therapy

## Abstract

Radiotherapy (RT) is one of the main clinical therapeutic strategies against cancer. Currently, multiple radiosensitizers aimed at enhancing X-ray absorption in cancer tissues have been developed, while limitations still exist for their further applications, such as poor cellular uptake, hypoxia-induced radioresistance, and unavoidable damage to adjacent normal body tissues. In order to address these problems, a cell-penetrating TAT peptide (YGRKKRRQRRRC)-modified nanohybrid was constructed by doping high-Z element Au in hollow semiconductor Cu_2−x_Se nanoparticles for combined RT and photothermal therapy (PTT) against breast cancer. The obtained Cu_2−x_Se nanoparticles possessed excellent radiosensitizing properties based on their particular band structures, and high photothermal conversion efficiency beneficial for tumor ablation and promoting RT efficacy. Further doping high-Z element Au deposited more high-energy radiation for better radiosensitizing performance. Conjugation of TAT peptides outside the constructed Cu_2−x_Se/Au nanoparticles facilitated their cellular uptake, thus reducing overdosage-induced side effects. This prepared multifunctional nanohybrid showed powerful suppression effects towards breast cancer, both in vitro and in vivo via integrating enhanced cell penetration and uptake, and combined RT/PTT strategies.

## 1. Introduction

Cancer has been the chief reason of death worldwide. Among primary therapeutic strategies, radiotherapy (RT) is a powerful one used for over a century, which noninvasively exerts tumoricidal effects by damaging DNA directly and generating cytotoxic reactive oxygen species (ROS) indirectly [1,2]. However, X-ray irradiation alone faces multiple challenges, such as the inevitable harms to neighboring normal tissues and radioresistance induced by tumor hypoxia [3,4]. Various radiosensitizers aimed at enhancing X-ray absorption in tissues have been developed, and it is essential for them to establish the properties of stability, good biocompatibility, and tumor targeted delivery and uptake for effective clinical practice. With the development of nanotechnology, nanoparticle radiosensitizers provide versatile strategies in order to deal with these problems [5]. Among them, high-Z elements (e.g., Au [6,7], Hf [8,9], Bi [10,11], and Gd [12])-containing nanoparticles can effectively absorb high-energy radiation, and therefore can be employed as radiosensitizers to heighten the response of tumor cells to radiation. Furthermore, semiconductor nanoparticles, such as Bi_2_Se_3_ [13], WS_2_ [14], Bi_2_S_3_ [15], and Cu_2−x_Se [16], have also been extensively explored as radiosensitizers and can generate ROS based on their particular band structures.

Another essential treatment strategy for improving RT efficacy is combining useful therapies with RT for synergistic effects [1]. PTT has been reported to be one of the most effective candidates for RT combination therapeutics [17,18]. By using photothermal agents absorbed in the near-infrared (NIR) range, PTT can transform NIR luminous energy into thermal energy for tumor ablation [19]. On one hand, hyperthermia induced during PTT can elevate radiosensitivity via increasing intratumoral blood flow, and thus enhance the oxygenation status and reverse hypoxia-induced radioresistance [20]. Additionally, hyperthermia results in double strand DNA breaks, preventing DNA repair after RT [21]. Furthermore, hyperthermia can kill radiation-insensitive cancer cells of the S-phase to raise the productivity of radiotherapy [22]. On the other hand, RT can compensate for the incomplete ablation of deep-seated tumors by NIR light, owing to its higher penetration ability [23]. Therefore, nanomaterials with both photothermal ablation and radiosensitizing properties are promising for improving therapeutic outcome.

Semiconductor Cu_2−x_Se (0 ≤ x ≤ 1) nanoparticles are ideal in RT and PTT combination therapeutics because of their radiosensitizing properties and high photothermal conversion efficiency acting as PTT agents [16,24,25,26]. Furthermore, the role of Cu and Se as important human trace elements confers Cu_2−x_Se nanoparticles satisfactory biocompatibility [27,28]. Chemically doping high-Z element Au in Cu_2−x_Se nanoparticles could deposit more energy for better radiosensitizing performance. Thus, we considered applying nanohybrids of Cu_2−x_Se/Au nanoparticles, hoping to acquire the synergistic effects of radiosensitizing and photothermal ablation.

As mentioned above, except for elevating radiosensitivity, ensuring the efficient delivery and cellular uptake in vivo is also vital for exerting their anticancer effects and preventing possible harms to normal tissues. Cell-penetrating peptides (CPPs), one of the most effective positive-charged modifications on nanomaterials, have been widely used in cell membrane infiltration and cellular internalization based on the electrostatic adsorption capacity of negative-charged cancer cell membranes [29,30]. Among them, TAT peptide is a classic one that can easily diffuse across cell membranes [31]. Our previous study modified TAT on surface of nanoparticles for breast cancer treatment, and in vitro/in vivo outcomes certificated evidently increased cellular internalization and nuclear localization mediated by TAT peptide [32]. Therefore, it is promising to design Cu_2−x_Se/Au nanoparticles with TAT peptide to enhance cellular uptake and avoid harmful radiation dosage for the highest therapeutic efficacy. 

Herein, TAT-modified Cu_2−x_Se/Au nanoparticles (Cu_2−x_Se/Au-TAT nanoparticles) with uniform size and morphology were developed, exhibiting satisfactory radiosensitizing effect and photothermal performance suitable for cancer RT/PTT combination therapy. As illustrated in Figure 1A, hollow Cu_2−x_Se nanoparticles were prepared via the sacrificial template method by using Cu_2_O nanoparticles as the sacrificial templates, and Cu_2−x_Se/Au nanoparticles were synthesized via in situ growing Au on Cu_2−x_Se nanoparticles. To strengthen the cellular uptake ability and therapeutic effects, TAT peptide was used to modify Cu_2−x_Se/Au nanoparticles. The combined RT/PTT effects of Cu_2−x_Se/Au-TAT nanoparticles against breast cancer are proposed in Figure 1B. After intratumoral injection, Cu_2−x_Se/Au-TAT nanoparticles can be taken up by breast cancer cells efficiently mediated by TAT peptide. Upon NIR laser irradiation, Cu_2−x_Se nanoparticles exert photothermal performance to ablate tumor via hyperthermia. Meanwhile, increased intratumoral blood flow induced by hyperthermia leads to improved oxygenation level, thus heightening the sensitivity of hypoxic tumor cells to RT. After exposure to X-ray irradiation, abundant high-Z element Au in Cu_2−x_Se/Au-TAT nanoparticles can deposit higher radiation energy within tumor to damage DNA, at the same time aiding Cu_2−x_Se nanoparticles to promote the generation of more cytotoxic ROS. Moreover, the constructed hollow nanostructure holds the potential to act as carriers for multifunctional agents, such as chemotherapeutic drugs [33], photosensitizer [34], photothermal agents [35], and so on, further expanding its application. Thus, this work highlights an excellent multifunctional nanoplatform based on Cu_2−x_Se/Au-TAT nanoparticles for combined RT/PTT therapy.

## 2. Results and Discussion

### 2.1. Synthesis and Characterization of Cu_2−x_Se/Au-TAT Nanoparticles

The synthesis process of Cu_2−x_Se/Au-TAT nanoparticles is shown in Figure 1A. Cu_2−x_Se/Au-TAT nanoparticles were synthesized via a multistep synthesis process. Briefly, in the presence of polyvinylpyrrolidone (PVP), Cu_2_O nanoparticles were synthesized by reducing copper chloride dihydrate (CuCl_2_⋅2H_2_O) with hydrazine hydrate (N_2_H_4_⋅H_2_O). Then, Cu_2−x_Se nanoparticles were prepared by a sacrificial template method with Cu_2_O nanoparticles as the sacrificial template and selenium powder as the selenium source. Afterward, Cu_2−x_Se/Au nanoparticles were prepared through a spontaneous redox process using hydrogen tetrachloroaurate (HAuCl_4_) as the source of Au. Finally, TAT was modified on the surface to improve the cellular uptake. Transmission electron microscopy (TEM) was employed to study the characteristics of size and morphology of the nanomaterials. As illustrated in Figure 1A, Cu_2_O nanoparticles were solid spheroids with an average size of 95 ± 5 nm, while Cu_2−x_Se nanoparticles maintained the spherical morphology with a hollow structure (136 ± 3 nm) (Figure 1B). After in situ deposition of Au on the surface of Cu_2−x_Se nanoparticles, the average particle sizes of Cu_2−x_Se/Au nanoparticles (154 ± 6 nm) increased compared with Cu_2−x_Se nanoparticle (Figure 1C). After surface modification with TAT, the generated Cu_2−x_Se/Au-TAT nanoparticles exhibited increased surface roughness (Figure 1D). The composition of nanoparticles was further verified by elemental mapping. From Figure 1E, it is clearly displayed that Cu_2−x_Se/Au nanoparticles contained Cu, Se, and Au, and the Au element was evenly distributed on the Cu_2−x_Se nanoparticles. According to the dynamic light scattering (DLS) assessment, compared with the particle size of Cu_2−x_Se/Au, the size of the final Cu_2−x_Se/Au-TAT increased slightly to 154 ± 6 nm (Figure 1F). As illustrated in Figure 1G, the zeta potentials of Cu_2_O, Cu_2−x_Se, and Cu_2−x_Se/Au nanoparticles were −9.3 ± 0.6, −18.4 ± 0.8, and −12.9 ± 0.7 mV, respectively, while the zeta potential of Cu_2−x_Se/Au-TAT nanoparticles was reversed to +25.1 ± 1.8 mV after surface modification with TAT. The results further clarified the successful introduction of TAT on Cu_2−x_Se/Au nanoparticles. The stability of Cu_2−x_Se/Au-TAT nanoparticles in aqueous solution was also detected by DLS, as demonstrated in Figure 1H; the minor variation in the hydrodynamic size in 5 days indicated Cu_2−x_Se/Au-TAT nanoparticles had a good stability.

UV−vis−NIR spectra was utilized to study the optical features of different nanoparticles. Figure 1I shows that the strong localized surface plasmon resonance (LSPR) of Cu_2−x_Se nanomaterials was weakened after the formation of Cu_2−x_Se/Au nanoparticles, and the absorption at about 560 nm that appeared in Cu_2−x_Se/Au nanoparticles was attributed to the LSPR of Au nanoparticles, which further proves that Au nanoparticles were effectively deposited on Cu_2−x_Se nanoparticles.

An X-ray photoelectron spectroscopy (XPS) analytical experiment was employed to study the valence states of elementary substances in Cu_2−x_Se nanoparticles and Cu_2−x_Se/Au nanoparticles (Figure 1J–L and Appendix A). As shown in Appendix A, the two strong peaks situated at 932.8 eV and 952.7 eV, accounting for 83%, corresponded to Cu 2p_3/2_ and Cu 2p_1/2_ of Cu (I), respectively, indicating that Cu (I) was the main valence state of Cu_2−x_Se. While the two weak peaks located at 934.4 eV and 953.9 eV, accounting for 17%, corresponded to Cu 2p_3/2_ and Cu 2p_1/2_ of Cu (II), accordingly, demonstrating that Cu (II) also accounted for a small part of Cu_2−x_Se. Moreover, the binding energy peak at 54.8 eV represents the characteristic Se 3d, corresponding to Se (II) (Appendix A). As shown in Figure 1J for Cu_2−x_Se/Au nanoparticles, the content of Cu (I) decreased to 61% and the content of Cu (II) increased to 39% compared with the content in Cu_2−x_Se nanoparticles, which proves that a part of Cu (I) was oxidized to Cu (II) during the generation process of Cu_2−x_Se/Au nanoparticles. From Figure 1K and Appendix A, it can be seen that the valence state of selenium does not change from Cu_2−x_Se to Cu_2−x_Se/Au. The binding energy of Au 4f_5/2_ and 4f_7/2_ located at 87.8 eV and 84.0 eV were attributed to Au (0), which proves that Au (III) was reduced to Au (0) (Figure 1L). Further, the compositions of Cu_2−x_Se nanoparticles and Cu_2−x_Se/Au nanoparticles were determined by an inductively coupled plasma optical emission spectrometer (ICP-OES). The Cu/Se atomic ratio in Cu_2−x_Se nanoparticles was 1.81:1, showing copper deficiency characteristics. The Cu/Se/Au atomic ratio in Cu_2−x_Se/Au nanoparticles is 0.48:1:0.51. The reduction of Cu/Se ratio in Cu_2−x_Se/Au nanoparticles further supported the reduction of Au (III) by Cu (I) in the XPS results. All of the above results clarify the spontaneous redox process from Cu_2−x_Se nanoparticles to Cu_2−x_Se/Au nanoparticles after the addition of HAuCl_4_ solution into Cu_2−x_Se nanoparticles.

### 2.2. Photothermal and Radiosensitizing Performances of Cu_2−x_Se/Au-TAT Nanoparticles

Cu_2−x_Se nanoparticles can be utilized as efficient photothermal agents thanks to the characteristic of strong LSPR in the region of NIR [16,24,25,26]. The photothermal features of Cu_2−x_Se/Au-TAT nanoparticles were researched by photothermal conversion tests. As visually illustrated in Figure 2A, after the solution of Cu_2−x_Se/Au-TAT nanoparticles (80 μg/mL) being performed with a 10-min 808 nm irradiation at 0.5, 1.0, and 1.5 W/cm^2^, the temperatures of the solutions increased to 45 °C, 53 °C, and 58 °C, respectively, which indicates that the increase in temperature had positive noteworthy correlation with irradiation dose. The increase in temperature in solutions of Cu_2−x_Se/Au-TAT nanoparticles also presented a concentration-dependent manner (Figure 2B). As visually displayed in Figure 2F, the temperature of Cu_2−x_Se/Au-TAT nanoparticles could increase to 53 °C under a 10-min irradiation at 1.0 W/cm^2^, while it was difficult for the water temperature to go up, showing that Cu_2−x_Se/Au-TAT nanoparticles had excellent photothermal properties. According to previous literature, NIR-triggered photothermal therapy usually requires hyperthermia of >50 °C to achieve the thorough ablation of tumor [36], while extremely higher temperatures could damage normal cells. For this reason, Cu_2−x_Se/Au-TAT nanoparticles at the concentration of 80 μg/mL under a 10-min laser irradiation at 1.0 W/cm^2^ were chose for the conditions of evaluating the subsequent photothermal ablation effects towards cancer cells. Moreover, the photothermal stability of Cu_2−x_Se/Au-TAT nanoparticles was studied by NIR laser irradiation repeatedly for 5 cycles (Figure 2C), and the decrease in peak value of temperature with the addition of cycle was ignorable, indicating excellent photothermal stability. According to the equations listed in the Materials and Methods section, the time constant (τ) was 328.96 s (Figure 2D,E), and the photothermal conversion efficiency (η) of Cu_2−x_Se/Au-TAT was approximately 64.6%, determined by a previously reported method [37]. The photothermal conversion efficiency of Cu_2−x_Se/Au-TAT is higher than some inorganic nanoparticles, such as CuO@AuCu-TPP (37.9%) [38], hollow Cu_2_Se nanozymes (50.9%) [39], and many organic nanoparticles, such as polypyrrole nanoparticles (22.6%) [40] and polydopamine-contained hydrogel (45.7%) [37].

Previous literatures had shown that the high-Z element and semiconductor hetero-nanostructures endow nanomaterials with better radiosensitization performance than high-Z element or semiconductor alone [26,41]. In order to verify the radiosensitization of Cu_2−x_Se/Au-TAT nanoparticles, ROS generation in solution was investigated by using a probe compound 2,7-dichlorodihydrofluorescein diacetate (DCFH-DA). As can be demonstrated from Figure 2G, the fluorescence intensity of the group of Cu_2−x_Se + RT was largely higher than control + RT, while the fluorescence intensities observed in the group of Cu_2−x_Se/Au + RT and Cu_2−x_Se/Au-TAT + RT were higher compared to those in the Cu_2−x_Se+RT group, which indicated that high-Z element Au and semiconductor Cu_2−x_Se both played important roles in promoting the production of ROS.

### 2.3. Biocompatibility and Cellular Uptake of Cu_2−x_Se/Au-TAT Nanoparticles

The cellular uptake of Cu_2−x_Se/Au-TAT nanoparticles in mouse triple-negative breast cancer cell line (4T1) were investigated firstly. As demonstrated in Figure 3A, compared with the free fluorescein isothiocyanate (FITC)-treated group, the obvious fluorescent signals have been observed in both FITC-labeled Cu_2−x_Se/Au-TAT- and Cu_2-x_Se/Au-treated groups, and the fluorescent signal in the FITC-labeled Cu_2−x_Se/Au-TAT-treated group was the strongest, indicating that the modification of TAT could enhance the uptake of Cu_2−x_Se/Au. Next, a 3-(4,5-dimethyl-2-thiazolyl)-2,5-diphenyl-2-H-tetrazolium bromide (MTT) experiment was employed to test the cytotoxicity of Cu_2−x_Se/Au-TAT nanoparticles in 4T1 and human umbilical vein endothelial cell lines (HUVECs). As can be seen from Figure 3B, the cell survival rate of 4T1 and HUVECs cells were still above 80%, though the concentration of Cu_2−x_Se/Au-TAT nanoparticles was raised to 120 µg/mL, which demonstrates that the nanoparticles had little cytotoxicity within this concentration range, which lays a good foundation for further experiments.

### 2.4. Photothermal Ablation and Radiosensitization Effects In Vitro

The photothermal behavior and radiosensitization effects of Cu_2−x_Se/Au-TAT nanoparticles were further investigated on 4T1 cells. As can been seen in Figure 3C, the antitumor efficacy of Cu_2−x_Se/Au-TAT nanoparticles demonstrated an obvious concentration-dependent way after carrying out an irradiation by 808 nm light. The cell survival rate reduced with the rise of concentrations of Cu_2−x_Se/Au-TAT nanoparticles, and the survival rate was only 50% after being cultured with 60 μg/mL Cu_2−x_Se/Au-TAT nanoparticles at 1.0 W/cm^2^ or 50 μg/mL at 1.5 W/cm^2^. Cell viability even decreased to 80% at 1.0 W/cm^2^ or 88% at 1.5 W/cm^2^ at 80 μg/mL. The above results demonstrate that Cu_2−x_Se/Au-TAT nanoparticles exhibited excellent photothermal conversion properties and could be used to ablate tumor cells in vitro.

Figure 2E has already shown that Cu_2−x_Se, Cu_2−x_Se/Au, and Cu_2−x_Se/Au-TAT nanoparticles could all produce ROS after X-ray irradiation, indicating that these nanoparticles could act as radiosensitizers for the purpose of radiotherapy. In order to further verify the radio-enhancement effect, a plate colony-formation experiment was conducted; 4T1 cells were co-culture with Cu_2−x_Se, Cu_2−x_Se/Au, and Cu_2−x_Se/Au-TAT nanoparticles and were then irradiated at a dose of 0, 2, 4, and 6 Gy, respectively. As Figure 3D shows, with the increase in X-ray dose, the cell survival rate of each group showed a downward trend. However, there was a great diversity in the cell livability among different groups (Figure 3E). The survival rate S(Cu_2−x_Se/Au-TAT) < S(Cu_2−x_Se/Au) < S(Cu_2−x_Se) < S(control), and the calculated sensitization enhancement ratio (SER) values of Cu_2−x_Se, Cu_2−x_Se/Au, and Cu_2−x_Se/Au-TAT were 1.76, 2.13, and 2.55, respectively. The cell livability of Cu_2−x_Se/Au nanoparticles was lower than that of Cu_2-x_Se nanoparticles, indicating the radio-enhancement effect by combining Cu_2−x_Se and Au. Cu_2−x_Se/Au-TAT nanoparticles had a higher therapeutic effect than Cu_2−x_Se/Au nanoparticles, perhaps owing to the higher cellular uptake caused by TAT.

### 2.5. PTT/RT Combination Therapy In Vitro

The above results have proven that Cu_2−x_Se/Au-TAT nanoparticles exerted the potential of not only killing tumor cells as a radiosensitizer, but also ablating tumor cells as a photothermal agent. As monotherapy of RT or PTT is not effective in eradicating tumors [42,43], the combination strategy of RT and PTT is a more desirable tool to enhance the treatment efficacy. Thus, we further studied the combined RT/PTT antitumor effect of Cu_2−x_Se/Au-TAT nanoparticles in vitro.

ROS generation was tested by utilizing the probe of DCFH-DA. As demonstrated in Figure 4A, obvious higher green fluorescence intensity was found in the groups of Cu_2−x_Se/Au-TAT or Cu_2−x_Se/Au treated with X-ray and/or NIR, compared with other groups, indicating the potential of Cu_2−x_Se/Au as an effective radiosensitizer to raise RT efficacy. The intensity of green fluorescence in the set treated with Cu_2−x_Se/Au-TAT under X-ray and NIR was evidently higher than the group treated with Cu_2−x_Se/Au-TAT under only X-ray, which may be attributed to the hyperthermia-induced cellular overactive state. When compared with the group treated with Cu_2−x_Se/Au followed by NIR and X-ray, the fluorescence intensity of the group treated with Cu_2−x_Se/Au-TAT followed by NIR and X-ray was slightly higher, which may be due to the improvement of cellular uptake by TAT.

Furthermore, DNA damage evaluation was conducted by γ-H_2_AX foci detection through confocal microscopy. As can be seen from Figure 4B, the group of Cu_2−x_Se/Au-TAT + PTT + RT showed the highest γ-H_2_AX fluorescent spots, which was obviously higher than the group of Cu_2−x_Se/Au-TAT+RT. The results of DNA damage were consistent with those of ROS generation detection, which further confirms that all three variables including RT, PTT, and TAT could promote ROS generation, thus resulting in DNA damage in the nucleus.

A cell colony formation assay, which is considered as the “gold standard” for assessment of cancer cells’ response to X-ray irradiation, was also carried out [15]. The results of clonogenic assay and clonogenic survival assay are shown in Figure 4C,D. The cell survival fraction reduced to 50% upon the treatment of Cu_2−x_Se/Au-TAT + PTT, 29% of Cu_2−x_Se/Au-TAT + RT, and 4% of Cu_2−x_Se/Au-TAT + PTT + RT, indicating that combining PTT and RT had the best therapeutic effect. In addition to the radiosensitization and photothermal influence of Cu_2−x_Se/Au, the presence of TAT led to the raised cellular uptake of nanoparticles and enhancement of the therapeutic effect.

The PTT/RT combination therapy efficacy of Cu_2−x_Se/Au-TAT nanoparticles was ulteriorly visually identified by calcein-AM/PI double staining. After receiving various treatments, 4T1 cells presented different intensities of green (referring to live cells) and red (referring to dead cells) fluorescence. As can been seen from Figure 4E, barely no dead cells were found in the control, Cu_2−x_Se/Au, and Cu_2−x_Se/Au-TAT groups without any treatment, or NIR/X-ray-irradiated control groups, while more dead cells were seen in the Cu_2−x_Se/Au and Cu_2−x_Se/Au-TAT groups treated with NIR or X-ray. Notably, almost all tumor cells were dead in Cu_2−x_Se/Au and Cu_2−x_Se/Au-TAT treated with NIR and X-ray, clearly indicating the Cu_2−x_Se/Au-TAT-induced combined PTT and RT effects.

### 2.6. PTT/RT Combination Efficacy of Cu_2−x_Se/Au-TAT In Vivo

The photothermal performance of Cu_2−x_Se/Au-TAT nanoparticles in vivo was investigated by irradiating the tumor areas of 4T1 tumor mice using an 808 nm laser after intratumoral injection of PBS and Cu_2−x_Se/Au-TAT nanoparticles, and the temperature variation was recorded by an NIR camera. As illustrated in Figure 5A,B, after the tumors underwent a 10-min irradiation at 1.5 W/cm^2^, the temperature of the tumor area with the treatment of Cu_2−x_Se/Au-TAT nanoparticles rose to about 51.3 °C, while the group injected with PBS only increased to about 39.3 °C. The results show that Cu_2−x_Se/Au-TAT nanoparticles had excellent photothermal efficacy in vivo.

Furthermore, the PTT and RT combination therapeutic effects of Cu_2−x_Se/Au-TAT nanoparticles were explored in vivo. After all treatments, the mice were put to death and tumors were obtained (Figure 6A). The tumor volume curves are shown in Figure 6B,C. No significant differences could be detected in the groups of control, Cu_2−x_Se/Au, Cu_2−x_Se/Au-TAT, and control + PTT, except for a slight decrease in tumor in the control + RT group. Mice treated with Cu_2−x_Se/Au-TAT + RT and Cu_2−x_Se/Au-TAT + PTT had an inhibitory effect on tumor volume to 70.3% and 86.3%, respectively, when compared to the control group. Tumor volume from mice receiving Cu_2−x_Se/Au or Cu_2−x_Se/Au-TAT mediated PTT alone decreased rapidly in the early stage of treatment, but increased gradually in the later stage, demonstrating that PTT alone could not completely ablate the tumor. Meanwhile, when combining PTT and RT, the inhibitory effect of Cu_2−x_Se/Au-TAT could reach 99.6%, indicating that combined PTT and RT could effectively achieve tumor ablation. Notably, the inhibitory effects of the Cu_2−x_Se/Au-TAT-related groups were slightly better than those of the Cu_2−x_Se/Au-related groups, which may be due to the better tissue permeability. To further estimate the therapeutic effect of Cu_2−x_Se/Au-TAT-enhanced PTT/RT on tumors, tumor tissue sections were carried out by staining with hematoxylin and eosin (H&E) and antibodies against γ-H_2_AX (Figure 6F,G). Compared with the control and the groups of Cu_2−x_Se/Au and Cu_2−x_Se/Au-TAT, obvious tumor nucleus fragmentation and nuclear shrinkage were observed in the groups of Cu_2−x_Se/Au-TAT + PTT and Cu_2−x_Se/Au-TAT + RT, and this situation was further enhanced when receiving both PTT and RT. In the meantime, the group of Cu_2−x_Se/Au-TAT + PTT + RT dramatically upregulated the expression levels of γ-H_2_AX, indicating that serious DNA damage was induced by PTT/RT combined treatment.

In order to evaluate the potential by-effects of these nanomaterials and the curative treatments, the chief organs of the mice, which consisted of heart, liver, spleen, and kidney, were acquired for H&E staining. No significant tissue injury and adverse effects on the main organs of mice were observed (Appendix A). Simultaneously, no obvious abnormalities were found in the body weight of the Cu_2−x_Se/Au-TAT-treated mice (Figure 6D,E). These results indicate the high biocompatibility of Cu_2−x_Se/Au-TAT nanoparticles within tested days.

## 3. Materials and Methods

### 3.1. Materials

CuCl_2_⋅2H_2_O was supplied by Kemiou Chemical Reagent (Tianjin, China). Sodium hydroxide (NaOH) was bought from Chemical Reagent Wholesale Company (Tianjin, China). N_2_H_4_⋅H_2_O was furnished by Taimo Biological Technology Co., Ltd. (Tianjin, China). PVP K30 (MW = 40,000) was procured from Shengpei New Material Co., Ltd. (Shanghai, China). Selenium powder (Se, ∼100 mesh, ≥99.5%) and sodium borohydride (NaBH_4_) were supplied by CSI Biochemical Technology Co., Ltd. (Tianjin, China). HAuCl_4_ was supplied by Fengchuan Chemical Reagent Co., Ltd. (Tianjin, China). TAT peptide was procured from China Peptides (99.8%, Shanghai, China). FITC and calcein-AM/PI double staining kit were bought from Meilun Biotechnology Co., Ltd. (Dalian, China). MTT, DCFH-DA, and 4′,6′-diamidino-2-phylindole (DAPI) were offered by Sigma-Aldrich (St. Louis, MO, USA). Hoechst 33258 was obtained from Invitrogen (Carlsbad, CA, USA). Giemsa stain was supplied by Tianjin Solomon Biotechnology Co., Ltd. (Tianjin, China). Monoclonal antibodies against γ-H_2_AX were procured from Abcam (Cambridge, MA, USA). Alexa-Fluor488-labeled goat anti-rabbit secondary antibody was provided by Biyuntian Biological Reagent Co., Ltd. (Shanghai, China). 

### 3.2. Cell Lines and Animals

4T1 and HUVECs were furnished by Biovector NTCC (Beijing, China). All cells were cultured with 10% FBS and 1% penicillin/streptomycin with 5% CO_2_ at 37 °C. Female BALB/c mice (4–6 weeks old) were supplied by SPF Biotechnology Co., Ltd. (Beijing, China). The mouse breast cancer xenograft model was developed through 100 μL subcutaneous injection of sterile phosphate buffer saline (PBS) (including 1.0 × 10^6^ 4T1 cells) on the right back of each mouse. When the tumor volume reached ∼100 mm^3^, the mice could be used for the following in vivo experiments. All animal research was approved by the Animal Ethics Committee of Tianjin Medical University, and all procedures were carried out conforming with the Guide for the Care and Use of Laboratory Animals.

### 3.3. Synthesis of Cu_2−x_Se Nanoparticles, Cu_2−x_Se/Au Nanoparticles, and Cu_2−x_Se/Au-TAT Nanoparticles

Hollow Cu_2−x_Se nanoparticles were prepared in reference to former methods, with minor modifications [44]. Briefly, 0.1 mmol CuCl_2_⋅2H_2_O was directly dissolved in a mixture of 58 mL 2-propanol and 2 mL water, following the addition of 0.2 g PVP and 0.24 mmol of NaOH solution in 1.2 mL 2-propanol. Next, the solution was reacted for 20 min. After that, 0.2 mL N_2_H_4_•H_2_O (35 wt%) was further added at a very slow speed, and the solution was stirred for another 10 min. Orange Cu_2_O sediments were immediately apparent upon adding N_2_H_4_•H_2_O. The obtained Cu_2_O nanoparticles were gained by centrifugation and subsequent water washing. 

In the preparation of hollow Cu_2−x_Se nanoparticles, 0.1 mmol of Se power and 0.3 mmol of NaBH_4_ were first dissolved in 20 mL water under the protection of N_2_, and the mixed solution was stirred for 30 min to acquire the Se^2-^ solution. Then, the above-mentioned double portion of Cu_2_O nanoparticles was added to the resultant Se^2-^ suspension and stirred for 3 h. The obtained Cu_2-x_Se nanoparticles were gained by centrifugation and subsequent water washing.

Next, 0.05 M of HAuCl_4_ solution was placed into the Cu_2−x_Se nanoparticles dropwise with a mole ratio of 1:0.375 (Cu_2−x_Se: HAuCl_4_) to prepare the Cu_2−x_Se/Au nanoparticles. The synthetic products were gained by centrifugation and redispersed in water.

To synthesize Cu_2−x_Se/Au-TAT nanoparticles, TAT solution was added to the above synthesized Cu_2−x_Se/Au solution dropwise refering to a mass ratio of Cu_2−x_Se/Au:TAT of 1:1.5, followed by stirring the mixture for 1 h. Finally, Cu_2−x_Se/Au-TAT nanoparticles were gained by centrifugation and subsequent water washing.

### 3.4. Characterization

The morphological features of the obtained samples were observed through a HT7700 TEM (Hitachi, Tokyo, Japan). The elemental mapping was recorded by a JEM-2100F instrument (JEOL, Tokyo, Japan). The absorbance of UV-vis-NIR was obtained by employing a UV-1510 spectrophotometer (Thermo Fisher Scientific, Waltham, MA, USA). The fluorescence spectrum was acquired by an F-7000 fluorescence spectrophotometer (Hitachi, Tokyo, Japan). The hydrodynamic size (Dh) and zeta potentials of the samples were obtained by a ZS90 Zetasizer (Malvern, Malvern, UK) (at Cu_2−x_Se concentration of 40 μg/mL. XPS was employed to study the valence states of elementary substances by utilizing an ESCALab 250XiSigma Probe instrument (Thermo Fisher Scientific, Waltham, MA, USA). Cu and Au content of the nanoparticles were determined by a Thermo Electron Corporation X7 ICP-OES (Thermo Fisher Scientific, Waltham, MA, USA). A Rad Source RS-2000 Pro (Rad Source Technologies, Atlanta, GA, USA) was performed for radiation. 

### 3.5. Photothermal Experiments

To estimate the photothermal performance of Cu_2−x_Se/Au-TAT nanoparticles, the aqueous suspensions of nanoparticles with multiple concentrations (0, 20, 40, 80, 120, 160 μg/mL) underwent irradiation with 808 nm light (1.0 W/cm^2^, 10 min). In addition, Cu_2−x_Se/Au-TAT nanoparticles (80 μg/mL) underwent irradiation for 10 min at various light powers (0.5, 1.0, and 1.5 W/cm^2^). The photothermal stability of Cu_2−x_Se/Au-TAT nanoparticles (80 μg/mL) was examined by laser ON/OFF assays for 5 cycles. For each cycle, the Cu_2−x_Se/Au-TAT aqueous solution underwent irradiation for 10 min with 808 nm light (1.0 W/cm^2^), following by light shut off to allow the solution to cool down to ambient temperature. The temperatures of the samples were recorded by an FLIR infrared camera (Nashua, NH, USA).

The photothermal conversion efficiency (η) of Cu_2−x_Se/Au-TAT was calculated according to the following equations:(1)$$\eta =\frac{\mathrm{hs}\left(T_{\max}-T_{\mathrm{sur}}\right)-Q_{\mathrm{dis}}}{I\left(1-10^{-A}\right)}$$
(2)$$\mathrm{hs}=\frac{\mathrm{mc}}{\tau }=\frac{\mathrm{mcLn}\left(\theta \right)}{-t}$$
(3)$$\theta =\frac{\left(T-T_{\mathrm{sur}}\right)}{\left(T_{\max}-T_{\mathrm{sur}}\right)}$$
where h is the heat transfer coefficient, s is the surface area of the container, Tmax is the maximum temperature of the Cu_2−x_Se/Au-TAT solution, Tsur is the maximum temperature of the surroundings, Qdis is the quantity of heat generated when the reagent is water, I is the laser power, A is the absorbance of the Cu_2−x_Se/Au-TAT solution at 808 nm, m is the mass of the water solution of nanoparticles, c is the specific heat capacity of the water (4.2 J/g), τ is the time constant of the sample, and T and t are the temperature of the sample and the time of the sample during the cooling process, respectively.

### 3.6. Detection of ROS

ROS detection was carried out by employing DCFH-DA as the probe. Firstly, 20 μL of DMSO solution containing DCFH-DA (0.02 M) and 2 mL of sodium hydroxide (0.01 M) were reacted in the absence of light at ambient temperature for 30 min. Afterwards, 18 mL of phosphate buffer (pH 7.2) was added into the above solution to stop the reaction. Next, DCFH was added into the sample solutions of all the groups. Eight groups were set: Control, Cu_2−x_Se, Cu_2−x_Se/Au, Cu_2−x_Se/Au-TAT, Control + X-ray, Cu_2−x_Se + X-ray, Cu_2-x_Se/Au + X-ray, Cu_2−x_Se/Au-TAT + X-ray. The final concentrations of DCFH were set at 10 μM, the final concentrations of Cu_2−x_Se were set at 40 μg/mL, and X-ray was administered at 4 Gy. Afterwards, solutions were shaken on the shaker for 2 h. Finally, the fluorescence of the solutions was obtained using an F-7000 spectrofluorometer with an excited wavelength of 488 nm.

### 3.7. Cellular Uptake Experiment

Firstly, Cu_2−x_Se/Au-TAT or Cu_2−x_Se/Au was labelled with FITC by adding FITC (100 μg/mL) into Cu_2−x_Se/Au-TAT or Cu_2−x_Se/Au (1 mg/mL) and reacted for 24 h. The exceeded FITC was removed through washing with water. Next, 4T1 cells were planted into 12-well plates (1 × 10^5^ per well) and cultured at 37 °C for 24 h. Subsequently, 4T1 cells were incubated with the medium containing free FITC, FITC-labeled Cu_2−x_Se/Au-TAT nanoparticles, and Cu_2−x_Se/Au nanoparticles for 4 h. After cleaning three times with PBS, the cells were fastened for 10 min, employing 4% paraformaldehyde. Finally, the cell nucleus was stained by DAPI and imaged, employing confocal fluorescence microscopy.

### 3.8. In Vitro Cytotoxicity Study

The cell cytotoxicity of Cu_2−x_Se/Au-TAT nanoparticles towards 4T1 cells and HUVECs was evaluated using the MTT method. Cells were spread evenly in 96-well plates (5 × 10^3^ per well) and cultured for twenty-four hours. Then, cells were cocultured with the medium containing Cu_2−x_Se/Au-TAT nanoparticles (10, 20, 40, 60, 80, and 120 μg/mL, respectively). After 4-h incubation, the cells were cleaned twice, employing PBS, 100 μL of fresh culture was added, and they were then continuously incubated for 20 h. Subsequently, the above cells were incubated for 4 h absent from dark. The medium was then discarded and they continued to be incubated with 100 μL of 1 mg/mL MTT solution. Afterwards, each well of the medium was discarded, 150 μL of DMSO was further added, and the plates then underwent a 5-min shake. The absorbance of each well was detected at 490 nm, employing a microplate reader.

To evaluate the photothermal ablation effects towards cancer cells, cells were planted into 96-well plates (5 × 10^3^ per well) at 37 °C overnight and incubated with multiple concentrations of Cu_2−x_Se/Au-TAT nanoparticles (30, 40, 50, 60, 70, and 80 μg/mL), respectively. After 4 h, the cells underwent irradiation with 808 nm light at 1.0 or 1.5 W/cm^2^ for 10 min, were washed two times, employing PBS, and supplemented with 100 μL of fresh DMEM for another 20 h. The cell livability was performed via the standard MTT test.

### 3.9. In Vitro Clonogenic Assay

4T1 cells were spread evenly into 6-well plates with different densities (250, 500, 1000, and 2000 per well) and cultured for 24 h. Next, the culture containing Cu_2−x_Se, Cu_2−x_Se/Au, and Cu_2−x_Se/Au-TAT nanoparticles at Cu_2−x_Se concentration of 40 μg/mL was added and incubated for 4 h, and then underwent irradiation with X-ray at 0, 2, 4, and 6 Gy, respectively. After washing with PBS and refreshing medium, all the treated groups were further incubated for 7 d. Finally, the above cells were rinsed two times with PBS, underwent 15-min immobilization with methanol, and were further stained using Giemsa dye for 30 min. Colonies covering at least 50 cells were counted, and the survival fraction (SF) was computed by (surviving colonies)/(cells seeded × plating efficiency) to evaluate the effects of different treatments.

SER was defined with the following formulae:S = 1 − (1 − exp (−*D/D_0_*)) *^N^*(4)
where S represents the cell survival fraction at a certain X-ray dose, *D* is representative of the dose of the irradiated X-ray, *D_0_* acts as the average lethal dose of radiation representing the sensitivity of cells to X-ray, and *N* represents the cell’s self-healing ability. *D_0_* and *N* were acquired by SPSS. SER was calculated as follows:SER = *D_q_* (control group)/*D_q_* (treated group)(5)
*D_q_* = In(N) × *D_0_*(6)
where *D_q_* is representative of the quasi-threshold dose, which represents the ability to repair sublethal injury.

In the combination therapy experiment, 4T1 cell were inoculated into 96-well plates (3000 cells/well). The experiment was divided into 12 groups: (1) Control (PBS), (2) Cu_2−x_Se/Au, (3) Cu_2−x_Se/Au-TAT, (4) Control + PTT, (5) Cu_2−x_Se/Au + PTT, (6) Cu_2−x_Se/Au-TAT + PTT, (7) Control + RT, (8) Cu_2−x_Se/Au + RT, (9) Cu_2−x_Se/Au-TAT + RT, (10) Control + PTT + RT, (11) Cu_2−x_Se/Au + PTT + RT, (12) Cu_2−x_Se/Au-TAT + PTT + RT. After 4 h incubation, groups 4, 5, 6, 10, 11, and 12 underwent 10-min irradiation with 808 nm light at 1.5 W/cm^2^. Afterwards, groups 7, 8, 9, 10, 11, and 12 were exposed to X-ray at 4 Gy. The above cells with different treatments were then rinsed with PBS, trypsinized, and seeded into 6-well plates for incubation of 7 d. Finally, the cells were rinsed with PBS, underwent 15-min immobilization with methanol, were further stained using Giemsa dye for 30 min, and counted.

### 3.10. In Vitro Detection of ROS Generation

4T1 cells were inoculated on confocal dishes (8 × 10^4^ per well). Twelve groups were set as described previously. Firstly, cells were incubated with control, Cu_2−x_Se/Au, and Cu_2−x_Se/Au-TAT nanoparticles at a Cu_2−x_Se concentration of 40 μg/mL for 4 h. Cells were then washed three times with PBS and underwent a 20-min incubation with DCFH-DA (10 μM) and Hoechst 33,258 (10 μM) at 37 °C. After being washed three times with PBS and adding 1 mL of PBS, the experimental groups were treated with or without a 10-min NIR irradiation (808 nm, 1.5 W/cm^2^), followed by treatment with or without X-ray (4 Gy), respectively. Finally, a fluorescence microscope was employed to observe the ROS created within cells

### 3.11. In Vitro DNA Damage Evaluation

To assess DNA double-strand injury, 4T1 cells in logarithmic growth were spread evenly into 12-well plates (8 × 10^4^ per well) and cultured for 24 h. Groups were set the same as in the in vitro ROS detection tests. These cells were incubated with control, Cu_2−x_Se/Au, and Cu_2−x_Se/Au-TAT nanoparticles at a Cu_2−x_Se concentration of 40 μg/mL for 4 h, and then treatments were performed with or without a 10-min NIR irradiation (808 nm, 1.5 W/cm^2^) and/or X-ray (4 Gy). Afterwards, these cells were washed three times with PBS, and then fastened for 10 min by 4% paraformaldehyde. Afterwards, the cells underwent a 10-min permeation by 0.2% Triton X-100 and a 1-h blocking with 3% BSA. After that, the cells were incubated with anti-γ-H_2_AX monoclonal antibody (1:250) at 4 °C overnight, followed by a treatment of secondary Alexa-488-conjugated goat anti-rabbit antibody (1:300). Finally, the cell nuclei underwent a 10-min staining by DAPI (1:900) and were imaged, employing confocal fluorescence microscopy.

### 3.12. Live–Dead Cell-Staining Assay

4T1 cells were inoculated in 12-well plates (1 × 10^5^ per well) and incubated for 24 h. Then, these cells were incubated with control, Cu_2−x_Se/Au, and Cu_2−x_Se/Au-TAT nanoparticles at a Cu_2−x_Se concentration of 60 μg/mL for 4 h. Following the various treatments mentioned above, a calcein-AM/PI double staining kit was employed to stain for 30 min to make a distinction between the live cells (green) and the dead cells (red). Finally, a Zeiss fluorescence microscope (Jena, Germany) was employed to observe the acquired results.

### 3.13. In Vivo Antitumor Efficacy

The 4T1 tumor-bearing mice were distributed into twelve groups of 6 at random, as described above, as long as the tumor grew to 100 mm^3^. The twelve groups of mice were then intratumorally injected with 50 μL control, Cu_2−x_Se/Au (at a Cu_2−x_Se concentration of 4 mg/mL), and Cu_2−x_Se/Au-TAT nanoparticles (at a Cu_2−x_Se concentration of 4 mg/mL), respectively. The mice of different groups then underwent irradiation with or without a 10-min NIR light (808 nm, 1.5 W/cm^2^) and/or X-ray (6Gy). The volume of tumors and the mouse weight was noted every second day during the whole treatment period. The volume of tumors was measured referring to the following computing formula: volume = 1/2(width^2^ × length). All tumors were imaged on the 20th day after treatments.

### 3.14. Histology Analysis In Vivo

The chief organs, which consisted of heart, liver, spleen, and kidney, were acquired for histology analysis after various treatments. These removed organs were fastened with 4% paraformaldehyde and were further made into paraffin sections for H&E staining. Additionally, the tumors were obtained after treatments for 48 h, fastened by 4% paraformaldehyde, and further embedded in paraffin. Slices were stained by H&E and γ-H2AX antibody. All these images of tissue slices were imaged, employing a fluorescence microscope.

### 3.15. Statistical Analysis

All the above results were assessed via Student’s *t*-test or one-way ANOVA. The experimental results were given as mean ± standard deviation (SD). *p* < 0.05 was representative of statistical significance.

## 4. Conclusions

In conclusion, multifunctional Cu_2−x_Se/Au-TAT nanoparticles were prepared by a spontaneous redox process between hollow Cu_2-x_Se nanoparticles and HAuCl_4_, with subsequent surface modification with TAT. They exhibited excellent stability, enhanced tumor cell uptake, excellent radiosensitization, and outstanding photothermal properties. Both in vitro and in vivo, compared with PTT or RT monotherapy, combination therapy of PTT and RT achieved a significant combined anticancer effect. This study provided an efficient mentality for RT/PTT combination therapy against breast cancer.

## Data Availability

Written informed consent has been obtained from the patient(s) to publish this paper.

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
