# Peer review of "A Cell-Penetrating Peptide Modified Cu2−xSe/Au Nanohybrid with Enhanced Efficacy for Combined Radio-Photothermal Therapy"

_molecules, 2023, doi:10.3390/molecules28010423_

Round 1

Reviewer 1 Report

The authors carried out a large-scale work, as a result of which promising outputs were obtained, however, requiring some clarifications/improvements.

2 Results and Discussions

From the description of the methodology and the results, it is not clear how the analysis of hydrodynamic size and zeta potentials of particles was carried out. The method used by the authors involves the analysis of a colloidal system, but it is not indicated what concentration of particles was analyzed. Although this parameter can affect the behavior of particles in solution

The authors do not explain the choice of the dose of nanoparticles (80 μg/ml) for the analysis of photothermal properties.

The authors do not explain the choice of different concentrations: when characterizing nanoparticles - 20, 40, 80,120, 160; assessment of cytotoxicity - 10.20, 40, 60, 80, 120; photothermal ablation - 30, 40, 50, 60, 70 and 80 μg/mL; radio-enhancement effect, a plate colony-formation experiment - it is not at all clear what doses were used. Same for other in vitro and in vivo studies

There is no statistical analysis in Figure 3 (significant differences were not noted), respectively. it is not clear, especially at 3b, whether there was a concentration-dependent toxicity or not.

There is no discussion of the results.

What are the limitations of the approach proposed by the authors?

3. Materials and Methods

Point 3.9 is not clear. In Vitro Clonogenic Assay "4T1 cells were spread evenly into 6-well plates with different densities (250, 500, 1000, and 2000 per well) and cultured for 24 h. Next, the culture containing Cu2-xSe, Cu2-xSe/Au and Cu2-xSe/Au-TAT nanoparticles was added and incubated for 4 hours".

How is the "different densities 250, 500, 1000, and 2000 per well" reflected in the results?

What kind of culture is "culture containing Cu2-xSe, Cu2-xSe/Au and Cu2-xSe/Au-TAT nanoparticles"? What is the concentration of nanoparticles in it?

Author Response

Results and Discussions

1. From the description of the methodology and the results, it is not clear how the analysis of hydrodynamic size and zeta potentials of particles was carried out. The method used by the authors involves the analysis of a colloidal system, but it is not indicated what concentration of particles was analyzed. Although this parameter can affect the behavior of particles in solution.

Response: Thanks for your professional suggestion. The concentration of nanoparticles used for determining hydrodynamic size and zeta potentials is about 40 μg/mL, which has been added in experimental section (3.4. Characterization) in the revised manuscript.

2. The authors do not explain the choice of the dose of nanoparticles (80 μg/ml) for the analysis of photothermal properties.

Response: Thanks for your professional comment. According to previous literature, NIR-triggered photothermal therapy usually requires hyperthermia of >50 °C to achieve thorough ablation of tumor [1], and extremely higher temperatures could damage normal cells. For the above reasons, we chose 80 μg/mL for the analysis of photothermal properties. From the results, the temperature of Cu2-xSe/Au-TAT nanoparticles under this concentration could increase to 53 oC under a 10-min laser irradiation at 1.0 W/cm2. The explanation has been added in Line 194, Page 5 in the revised manuscript.

Reference:

  1. Yang, Y.; Zhu, W.; Dong, Z.; Chao, Y.; Xu, L.; Chen, M.; Liu, Z. 1D Coordination Polymer Nanofibers for Low-Temperature Photothermal Therapy. Mater. 2017, 1703588

3. The authors do not explain the choice of different concentrations: when characterizing nanoparticles - 20, 40, 80,120, 160; assessment of cytotoxicity – 10, 20, 40, 60, 80, 120; photothermal ablation - 30, 40, 50, 60, 70 and 80 μg/mL; radio-enhancement effect, a plate colony-formation experiment - it is not at all clear what doses were used. Same for other in vitro and in vivo studies

Response: Thank you for your careful review and professional suggestion.

In MTT experiment, the cell survival rate of 4T1 and HUVECs cells were above 80% at the concentration of Cu2-xSe/Au-TAT of 120 µg/mL. The concentration lower than 120 µg/mL could be used in subsequent cellular and animal studies.

In photothermal ablation experiment, the survival rate was only 50% with 60 µg/mL Cu2-xSe/Au-TAT nanoparticles at 1.0 W/cm2 or 50 µg/mL at 1.5 W/cm2. Cell viability had a decreasing of even 80% at 1.0 W/cm2 and 88% at 1.5 W/cm2 at 80 µg/mL. Therefore, the concentration lower than 80 µg/mL can be used for synergistic therapy, the concentration higher than 80 µg/mL was not appropriate.

In in vitro cell experiments, we made an exploration at the early stage of the project, and we found that the combined effect of photothermal therapy and radiotherapy can be seen at the concentration of 40 μg/mL, while almost all the cells were dead when treated with photothermal therapy at 80 μg/mL.

In in vivo experiments, the concentration of intratumoral injection was selected according to the data of photothermal heating in vivo. The temperature could be over 50 °C to achieve the thorough ablation of tumor [1] when the tumor was treated with nanoparticles combined PTT at this concentration.

In the revised manuscript, we have described the doses which we have used in the experiments.

Reference:

  1. Yang, Y.; Zhu, W.; Dong, Z.; Chao, Y.; Xu, L.; Chen, M.; Liu, Z. 1D Coordination Polymer Nanofibers for Low-Temperature Photothermal Therapy. Mater. 2017, 1703588

4. There is no statistical analysis in Figure 3 (significant differences were not noted), respectively. it is not clear, especially at 3b, whether there was a concentration-dependent toxicity or not.

Response: Thank you for your professional comment. Significant differences have been added in Figure 3C and Figure 3E. We also made statistical analysis of Figure 3B, while there were no significant differences.

5. There is no discussion of the results.

Response: Thanks for the reviewer’s kind suggestion. We found many articles published in Molecules put the results and discussions into one part, shown as follow. In our article, the results and discussions were highly relevant, and it is more appropriate to write these two parts together. Therefore, the relevant discussions of the results have been given in Section 2, Results and Discussion.

 Reference

  1. Cho, I.K.; Shim, M.K.; Um, W.; Kim, J-H.; Kim, K. Light-Activated Monomethyl Auristatin E Prodrug Nanoparticles for Combinational Photo-Chemotherapy of Pancreatic Cancer. Molecules 2022, 27, 2529.
  2. Xu, H.; Ling, J.; Zhao, H.; Xu, X.; Ouyang, X.-K.; Song, X. In vitro Antitumor Properties of Fucoidan-Coated, Doxorubicin-Loaded, Mesoporous Polydopamine Molecules 2022, 27, 8455.
  3. Lu, R.; Wang, W.; Dong, B.; Xu, C.; Li, B.; Sun, Y.; Liu, J.; Hong, B. Self-Assembled CuCo2S4 Nanoparticles for Efficient Chemo-Photothermal Therapy of Arterial Inflammation. Molecules 2022, 27, 8134.

6. What are the limitations of the approach proposed by the authors?

Response: Thank you for your careful review. Although there is no short-term acute toxicity, the long-term in vivo degradation of gold-containing nanoparticles remains a challenge.

7. Point 3.9 is not clear. In Vitro Clonogenic Assay "4T1 cells were spread evenly into 6-well plates with different densities (250, 500, 1000, and 2000 per well) and cultured for 24 h. Next, the culture containing Cu2-xSe, Cu2-xSe/Au and Cu2-xSe/Au-TAT nanoparticles was added and incubated for 4 hours". How is the "different densities 250, 500, 1000, and 2000 per well" reflected in the results? What kind of culture is "culture containing Cu2-xSe, Cu2-xSe/Au and Cu2-xSe/Au-TAT nanoparticles"? What is the concentration of nanoparticles in it?

Response: Thank you for your careful review. After cell digestion and counting, cell suspension with higher concentration was prepared, then the cell suspension was diluted to get different cell densities, as 250, 500, 1000, and 2000 cells per well. In cloning experiments, the number of cells inoculated varies with different doses of radiation, and the smaller the dose, the fewer cells should be placed. If not, too many clones are overlapping after seeding cells and therefore this well is not reliable for counting [1]. The culture containing Cu2-xSe, Cu2-xSe/Au and Cu2-xSe/Au-TAT nanoparticles was prepared by adding Cu2-xSe, Cu2-xSe/Au and Cu2-xSe/Au-TAT nanoparticles into DMEM full medium, and the concentration was set at Cu2-xSe concentration of 40 ug/mL, which had been added in 3.9. In Vitro Clonogenic Assay.

Reference

  1. Franken, N.A.; Rodermond, H.M.; Stap, J.; Haveman, Jaap, J.; van Bree, C. Clonogenic assay of cells in vitro. Protoc. 2006, 1(5):2315-2319.

Reviewer 2 Report

In this study, a cell-penetrating TAT peptide-modified nanohybrid was constructed by doping high-Z element Au in hollow semiconductor Cu2−xSe nanoparticles for combined RT and photothermal therapy (PTT) against breast cancer. The obtained Cu2−xSe nanoparticles possessed excellent radiosensitizing properties based on their particular band structures, and high photothermal conversion efficiency beneficial for tumor ablation and promoting RT efficacy. Further doping high-Z element Au deposited more high-energy radiation for better radiosensitizing performance. Conjugation of TAT peptide outside the constructed Cu2−xSe/Au nanoparticles facilitated their cellular uptake, thus reducing overdosage-induced side effects. This is an excellent works and can be accepted for publication in this journal after a minor revision.

1.     Figure 1i,j: N2 adsorption-desorption isotherm and pore size distributions of Cu2-xSe nanoparticles. I think these two images should be deleted from the work, since there was no specific application of these pores (like drug loading).

2.     How about the photothermal conversion efficiency of the Cu2−xSe/Au NPs? These point should be studied, and compared with other nano-platforms.

3.     I suggest the authors to define each abbreviation used in the text when it first appears, including in the abstract (like TAT); in addition, consider using fewer abbreviations in order to make the text easily accessible to a broad audience, only when absolutely necessary.

4.     Some closely related work on the tumor therapy biomedical applications of related bionanomaterials, such as polypyrrole, hydrogels (Oxidation triggered formation of polydopamine-modified carboxymethyl cellulose hydrogel for anti-recurrence of tumor; Enhanced photothermal and chemotherapy of pancreatic tumors by degrading the extracellular matrix). A comparison with them in the discussion part is highly welcome.

In general, this work seems to be quite interesting and I would like to see the revision if possible.

Author Response

1. Figure 1i,j: N2 adsorption-desorption isotherm and pore size distributions of Cu2-xSe nanoparticles. I think these two images should be deleted from the work, since there was no specific application of these pores (like drug loading).

Response: Thank you for the reviewer’s kind suggestion. Figure 1I,J and the description of N2 adsorption-desorption isotherm and pore size distributions in 2.1, 3.4 and Figure 1 have been deleted in the revised manuscript.

2. How about the photothermal conversion efficiency of the Cu2−xSe/Au NPs? These point should be studied, and compared with other nano-platforms.

Response: Thanks for the reviewer’s kind suggestion. The photothermal conversion efficiency of the Cu2−xSe/Au-TAT NPs had been calculated, and compared with other nano-platforms. The related discussions have been added in “2.2. Photothermal and radiosensitizing performances of Cu2-xSe/Au-TAT nanoparticles” in the revised manuscript.

3. I suggest the authors to define each abbreviation used in the text when it first appears, including in the abstract (like TAT); in addition, consider using fewer abbreviations in order to make the text easily accessible to a broad audience, only when absolutely necessary.

Response: Thank you for your careful review. We've detected all the abbreviations carefully. The words “TAT, PVP, CuCl2⋅2H2O, N2H4⋅H2O, HAuCl4, TEM, DLS, XPS, ICP-OES, DCFH-DA, 4T1, FITC, MTT, HUVECs, SER and H&E” which appeared for the first time in this article had been defined.

4. Some closely related work on the tumor therapy biomedical applications of related bionanomaterials, such as polypyrrole, hydrogels (Oxidation triggered formation of polydopamine-modified carboxymethyl cellulose hydrogel for anti-recurrence of tumor; Enhanced photothermal and chemotherapy of pancreatic tumors by degrading the extracellular matrix). A comparison with them in the discussion part is highly welcome.

Response: Thank you for providing the relevant literature. With this in mind, we compared the photothermal conversion efficiency of Cu2−xSe/Au-TAT NPs with that of different inorganic and organic nanoparticles (including polypyrrole and hydrogels). And the related discussions have been added in in “2.2. Photothermal and radiosensitizing performances of Cu2-xSe/Au-TAT nanoparticles” in the revised manuscript.

Round 2

Reviewer 1 Report

After correction, the manuscript improved significantly. I recommend accepting the manuscript for publication in the present form.